# Prognostic Significance of Glucose Metabolism as GLUT1 in Patients with Pulmonary Pleomorphic Carcinoma

**DOI:** 10.3390/jcm9020413

**Published:** 2020-02-03

**Authors:** Hisao Imai, Kyoichi Kaira, Hideki Endoh, Kazuyoshi Imaizumi, Yasuhiro Goto, Mitsuhiro Kamiyoshihara, Takayuki Kosaka, Toshiki Yajima, Yoichi Ohtaki, Takashi Osaki, Yoshihito Kogure, Shigebumi Tanaka, Atsushi Fujita, Tetsunari Oyama, Koichi Minato, Takayuki Asao, Ken Shirabe

**Affiliations:** 1Division of Respiratory Medicine, Gunma Prefectural Cancer Center, Ota 373-8550, Japan; m06701014@gunma-u.ac.jp (H.I.); kminato@gunma-cc.jp (K.M.); 2Department of Innovative Immune-Oncology Therapeutics, Gunma University Graduate School of Medicine, Maebashi 371-8511, Japan; yajimatos@yahoo.co.jp (T.Y.); kshirabe@gunma-u.ac.jp (K.S.); 3Department of Respiratory Medicine, Comprehensive Cancer Center, International Medical Center, Saitama University Hospital, Hidaka 350-1298, Japan; 4Department of Thoracic Surgery, Saku Central Hospital Advanced Care Center, Saku 385-0051, Japan; hidend0509@yahoo.co.jp; 5Department of Respiratory Medicine, Fujita Health University, Toyoake 470-1192, Japan; jeanluc@fujita-hu.ac.jp (K.I.); gotoyasu510@gmail.com (Y.G.); 6Department of General Thoracic Surgery, Japanese Red Cross Maebashi Hospital, Maebashi 371-0811, Japan; micha2005jp@yahoo.co.jp; 7Division of Thoracic Surgery, Takasaki General Medical Center, Takasaki 370-0829, Japan; tkosaka133@gmail.com; 8Department of General Surgical Science, Gunma University Graduate School of Medicine, Maebashi 371-8511, Japan; yohtakiadvanced@gmail.com; 9Department of Respiratory Medicine, Shibukawa Medical Center, Shibukawa 377-0280, Japan; tosaki@xb4.so-net.ne.jp; 10Department of Respiratory Medicine, Nagoya Medical Center, Nagoya 460-0001, Japan; yo-kogure@umin.ac.jp; 11Department of Respiratory Surgery, Isesaki Municipal Hospital, Isesaki 372-0817, Japan; tanakasigebumi@yahoo.co.jp; 12Division of Thoracic Surgery, Gunma Prefectural Cancer Center, Ota 373-8550, Japan; afujita@gunma-cc.jp; 13Department of Diagnostic Pathology, Gunma University Graduate School of Medicine, Maebashi 371-8511, Japan; oyama@gunma-u.ac.jp; 14Big Data Center for Integrative Analysis, Gunma University Initiative for Advance Research, Maebashi 371-8511, Japan; asao@gunma-u.ac.jp

**Keywords:** pulmonary pleomorphic carcinoma, prognostic factor, glucose transporter 1

## Abstract

Glucose metabolism is necessary for tumor progression, metastasis, and survival in various human cancers. Glucose transporter 1 (GLUT1), in particular, plays an important role in the mechanism of ¹⁸F-FDG (2-[¹⁸F]-fluoro-2-deoxy-d-glucose) within tumor cells. However, little is known about the clinicopathological significance of GLUT1 in patients with pulmonary pleomorphic carcinoma (PPC). Adenocarcinoma, squamous cell carcinoma, adenosquamous cell carcinoma, poorly differentiated carcinoma, large cell carcinoma, and others were identified as epithelial components, and spindle-cell type, giant-cell type, and both spindle- and giant-cell types were identified as sarcomatous components. This study was performed to determine the prognostic impact of GLUT1 expression in PPC. Patients with surgically resected PPC (n = 104) were evaluated by immunohistochemistry analysis to detect GLUT1 expression and determine the Ki-67 labeling index using specimens of the resected tumors. GLUT1 was highly expressed in 48% (50/104) of all patients, 42% (20/48) of the patients with an adenocarcinoma component, and 53% (30/56) of the patients with a nonadenocarcinoma component. High expression of GLUT1 was significantly associated with advanced stage, vascular invasion, pleural invasion, and tumor cell proliferation as determined by Ki-67 labeling. GLUT1 expression and tumor cell proliferation were significantly correlated according to the Ki-67 labeling in all patients (Spearman’s rank; r = 0.25, *p* < 0.01). In multivariate analysis, GLUT1 was identified as a significant independent marker for predicting a poor prognosis. GLUT1 is an independent prognostic factor for predicting the poor prognosis of patients with surgically resected PPC.

## 1. Introduction

Pulmonary pleomorphic carcinoma (PPC) is a rare disease with an incidence of 0.1%–0.4% among all lung cancers and shows a poor prognosis because of its resistance to systemic chemotherapy [1]. PPC includes carcinomatous and sarcomatoid components and is classified as a subtype of sarcomatoid carcinoma of the lung by the World Health Organization histologic classification of lung neoplasms [2,3]. Because of its rarity and low treatment efficacy, most patients with PPC exhibit recurrence even after complete surgical resection; moreover, there are no standard treatments for patients with advanced and inoperable PPC. The development of appropriate treatments and identification of predictive biomarkers are critical for improving the prognosis of patients with complex histologies, such as PPC.

Glucose metabolism is associated with tumor progression and metastases, and is used in molecular imaging, such as 2-[¹⁸F]-fluoro-2-deoxy-d-glucose (¹⁸F-FDG) positron emission tomography (PET), to detect cancers [4]. Although there are several types of glucose transporters (GLUTs), glucose transporter 1 (GLUT1) and GLUT3 are strongly expressed on the membrane of tumor cells, and a meta-analysis demonstrated GLUT1 to be a prognostic marker for predicting worse outcomes in patients with lung cancer [4]. ¹⁸F-FDG accumulates in tumor cells via GLUT1, a process closely associated with poor prognosis and tumor progression in patients with lung cancer [5]. We previously showed that ¹⁸F-FDG uptake in PPC is closely related to the presence of GLUT1 and angiogenesis, and that the accumulation of ¹⁸F-FDG and the expression level of GLUT1 were significantly higher in patients with PPC than those with other nonsmall cell lung cancer (NSCLC) [6]. This indicates that tumor glucose metabolism involving GLUT1 plays a crucial role in the carcinogenesis of PPC. ¹⁸F-FDG-PET can be used to detect primary and metastatic lesions for disease staging in patients with PPC. From a pathological perspective, studies are needed to determine how the expression of GLUT1 in cancer-specific glucose metabolism reflects the survival and metastasis of patients with PPC. However, little is known about the clinicopathological relevance of GLUT1 expression in patients with PPC.

In this clinicopathological study, we examined the prognostic role of GLUT1 expression in patients with surgically resected PPC.

## 2. Experimental Section

### 2.1. Patients

Between August 2001 and October 2015, 104 patients with histologically confirmed PPC who underwent surgical resection at multiple institutions were enrolled in this study. Pleomorphic carcinoma was diagnosed according to the 2015 World Health Organization Classification of Tumours [2]. Diagnoses were confirmed by light microscopy and immunohistochemistry. PPC was defined as NSCLC containing at least 10% sarcomatoid components. This study included 104 surgically resected primary tumors in accordance with institutional guidelines and the Declaration of Helsinki. The institutional review boards of all participating institutions approved this study. Mortality and recurrence were determined using medical records. The tumor samples were collected in our previous study [7,8,9].

### 2.2. Immunohistochemical Staining

GLUT1 expression was assessed by immunohistochemical staining using a rabbit anti-GLUT1 polyclonal antibody (Abcam, Cambridge, UK; 1:200 dilution). The reaction was visualized using the Histofine Simple Stain MAX-PO (Multi) Kit (Nichirei, Tokyo, Japan), according to the manufacturer’s instructions. The detailed protocol for immunostaining has been published elsewhere [4]. Negative controls were incubated without primary antibody, and no staining was observed. GLUT1 expression was considered positive only if distinct cytoplasmic and plasma membrane staining was present. GLUT1 expression was scored as follows: 1, ≤10% of tumor area stained; 2, 11%–25% stained; 3, 26%–50% stained; 4, 51%–75% stained; and 5, ≥76% stained. Tumors in which the stained tumor cells were scored ≥4 were considered as “high-expression” tumors.

Immunohistochemical staining for Ki-67 was performed as described previously [4] using a murine monoclonal antibody against Ki-67 (Dako, Glostrup, Denmark; 1:40 dilution). Highly cellular areas of the immunostained sections were assessed for Ki-67. All epithelial cells with nuclear staining of any intensity were defined as high-expression epithelial cells. Approximately 1000 nuclei were counted on each slide. Proliferative activity was assessed as the percentage of Ki-67-stained nuclei (Ki-67 labeling index) in the sample. The median Ki-67 labeling index value was evaluated, and tumor cells with greater than median Ki-67 labeling index value were defined as high-expression tumor cells. All sections were assessed by light microscopy in a blinded manner by at least two investigators. In case of discrepancies, both investigators evaluated the slides simultaneously until reaching a final consensus. Neither of the investigators had knowledge of the patient outcomes.

### 2.3. Statistical Analysis

Statistical analyses were performed using Student’s *t*- and χ^2^-tests for continuous and categorical variables, respectively. Correlations were analyzed using nonparametric Spearman’s rank tests. The Kaplan–Meier method was used to estimate survival as a function of time, and survival differences were analyzed by log-rank tests. Overall survival (OS) was defined as the time from tumor resection to death from any cause. Disease-free survival (DFS) was defined as the time between tumor resection and the first episode of disease progression or death. Univariate and multivariate survival analyses were performed using Cox proportional hazards models and a logistic regression model for radical surgery. *p* < 0.05 was considered to indicate statistical significance. All statistical analyses were performed using GraphPad Prism version 7 (GraphPad Software, San Diego, CA, USA) and JMP Pro version 14.0 (SAS Institute, Inc., Cary, NC, USA).

## 3. Results

### 3.1. Patient Demographics and Immunohistochemistry

GLUT1 expression was assessed in 104 patients (79 males, 25 females; median age 69 years, range 35–88 years) and correlated with patient’s clinical information. All patients were diagnosed using resected primary tumors. Histologic analysis revealed that 29 patients with PPC harbored a combination of carcinomatous and sarcomatous components. In the remaining 75 primary tumors, carcinomatous components were identified in 48 patients with adenocarcinoma, 13 with squamous cell carcinoma, 8 with adenosquamous cell carcinoma, 2 with poorly differentiated carcinoma, and 4 with Pe. Of the sarcomatous components, 69 patients exhibited spindle-cell type, 10 giant-cell type, and 25 both spindle- and giant-cell types. Each percentage of epithelial and sarcomatous components is shown in Appendix A. The day of surgery was considered the starting day for measuring postoperative survival. The median follow-up period was 476 days (range, 30–4519 days).

Patient demographics data according to GLUT1 expression are listed in Table 1. Immunohistochemical analyses were performed for 104 primary sites with PPC. GLUT1 was stained on the cell membranes of tumor specimens, and there was no evidence of normal tissue without red blood cells. Figure 1 shows the representative images of GLUT1 expression in patients with PPC. Figure 2 shows the distribution of GLUT1 expression according to a scoring system. The frequencies of scores 1, 2, 3 4, and 5 for GLUT1 were 11%, 3%, 25%, 32%, and 19%, respectively. The percentage of samples showing high GLUT1 expression was 48% (50/104). High expression of GLUT1 was found to be significantly associated with advanced stage, vascular invasion, pleural invasion, and tumor cell proliferation, as determined by the Ki-67 index. There was a significant correlation between GLUT1 expression and tumor cell proliferation according to the Ki-67 labeling index in all patients (Spearman’s rank; *r* = 0.25, *p* < 0.01).

Next, epithelial histological types such as adenocarcinoma (AC) and non-AC were assessed. No significant difference in the frequency of high GLUT1 expression was observed between patients with AC (20/48) and non-AC (30/56) (*p* = 0.24).

### 3.2. Univariate and Multivariate Survival Analysis

The median DFS and OS of all patients were 449 and 991 days, respectively. In the analysis according to the epithelial histology, the median DFS and OS of patients with AC and non-AC components were 522 and 1038 days and 336 and 507 days, respectively. In total, 60 patients died, and recurrence after initial surgery was observed in 59 patients. The above survival information has been previously described [7,8,9]. The results of the survival analysis are listed in Table 2. The Kaplan-Meier survival curve of all patients with high or low GLUT1 expression is shown in Figure 3. According to the univariate analysis, disease stage and GLUT1 were identified as significant factors for predicting worse OS after surgery and disease stage; pleural invasion and GLUT1 displayed a close association with poor DFS. The different variables with a cut-off of *p* < 0.05 were screened based on the results of the univariate log-rank test. In all patients, the disease stage and GLUT1 were confirmed as independent prognostic factors related to worse OS and DFS by multivariate analysis. Next, we analyzed the prognostic significance of GLUT1 expression according to the epithelial histological types of PPC (AC and non-AC component). Figure 3 shows the Kaplan-Meier survival curve of patients with AC and non-AC components. In univariate analysis, patients with a non-AC component with high GLUT1 expression showed a significantly worse OS and DFS than low GLUT1 expression compared to those with an AC component.

Figure 4 shows the forest plot of the one-year OS and DFS rates according to GLUT1 expression for each variable. Patients with high GLUT1 expression exhibited a worse OS and DFS than those with low GLUT1 expression for different variables except for stages III and IV.

## 4. Discussion

We examined the prognostic significance of GLUT1 expression in patients with surgically resected PPC. We found that overexpression of GLUT1 is an independent factor for predicting poor outcomes and is useful as a prognostic marker in patients with a non-AC component. In the patients with non-AC, OS and DFS showed the highest difference between low-GLUT1 and high-GLUT1 compared with all patients and subgroup patients with AC component. The value of GLUT1 as a prognostic marker differed according to the epithelial histology of PPC.

A previous meta-analysis of 1423 patients with lung cancer revealed a relationship between GLUT1 expression and clinicopathological parameters [5]. This study described that positive expression of GLUT1 was significantly associated with squamous cell carcinoma, poorly differentiated tumors, lymph node metastases, large tumor size, and advanced tumor stage. In the present study, no significant difference in the frequency of high GLUT1 expression was observed between patients with non-AC and AC components. In our analysis according to epithelial histology, however, high expression of GLUT1 was identified as a significant factor for predicting worse outcomes in patients with a non-AC component compared to those with an AC component. A previous study reported that the accumulation of FDG was closely linked to poor prognosis in patients with AC, indicating the prognostic role of glucose metabolism as a significant prognostic predictor [10]. Although the limited sample size may have biased our results, our study suggests that the role of GLUT1 as a prognostic predictor in the histology of the AC component differs between patients with PPC and NSCLC. Here, we demonstrated that high expression of GLUT1 was strongly correlated with advanced stage, vascular invasion, pleural invasion, and tumor cell proliferation. These findings correspond to those of a study on NSCLC [5].

Recently, we reported the prognostic significance of amino acid transporter 1 (LAT1) expression in patients with surgically resected NSCLC [7]. LAT1 was highly expressed in patients with PPC, and there was a close relationship between high LAT1 expression and a worse prognosis. In the analysis according to histological type, the expression of LAT1 was significantly lower in patients with an AC component than in those without an AC component; however, the role of LAT1 as a predictive marker related to poor prognosis did not differ between patients with AC and non-AC. This contradicts the findings of the current study.

There were several limitations to our study. First, the sample size was small because PPC is a rare entity, which may have biased the results. However, compared to previous studies, this was a large-scale investigation using tumor samples collected from multiple institutions. As it is difficult to definitively diagnose PPC using biopsy samples, we collected tumor samples from patients with surgically resected PPC. Second, the expression of GLUT1 has been shown to be closely associated with tumor progression, metastases, and survival of PPC; however, it remains unknown whether GLUT1 expression is correlated with the uptake of ¹⁸F-FDG within PPC tumor cells. Although a previous exploratory study indicated a close relationship between GLUT1 expression and ¹⁸F-FDG accumulation in patients with PPC, it is necessary to validate this correlation using different cohorts with more than 100 tumor samples. Finally, GLUT1 is found to be a targeting molecule for PPC; however, inhibition of glucose metabolism may be harmful to normal cells rather than cancer cells. Therefore, it may be difficult to administer inhibitors of GLUT1 as a treatment for PPC with a non-AC component in clinical practice. Further studies are needed to develop a selective inhibitor of GLUT1 to diminish the tumor growth and metastases of PPC.

## 5. Conclusions

GLUT1 is an independent predictor of poor prognosis in patients with surgically resected PPC, particularly in those with an AC component. Although GLUT1 is widely expressed in human cancers, tumor glucose metabolism was identified as an essential factor related to tumor cell proliferation, survival, and pathogenesis. Additional studies are needed to determine the therapeutic potential of GLUT1 inhibitors in patients with advanced PPC.

## Figures and Tables

**Figure 1 jcm-09-00413-f001:**
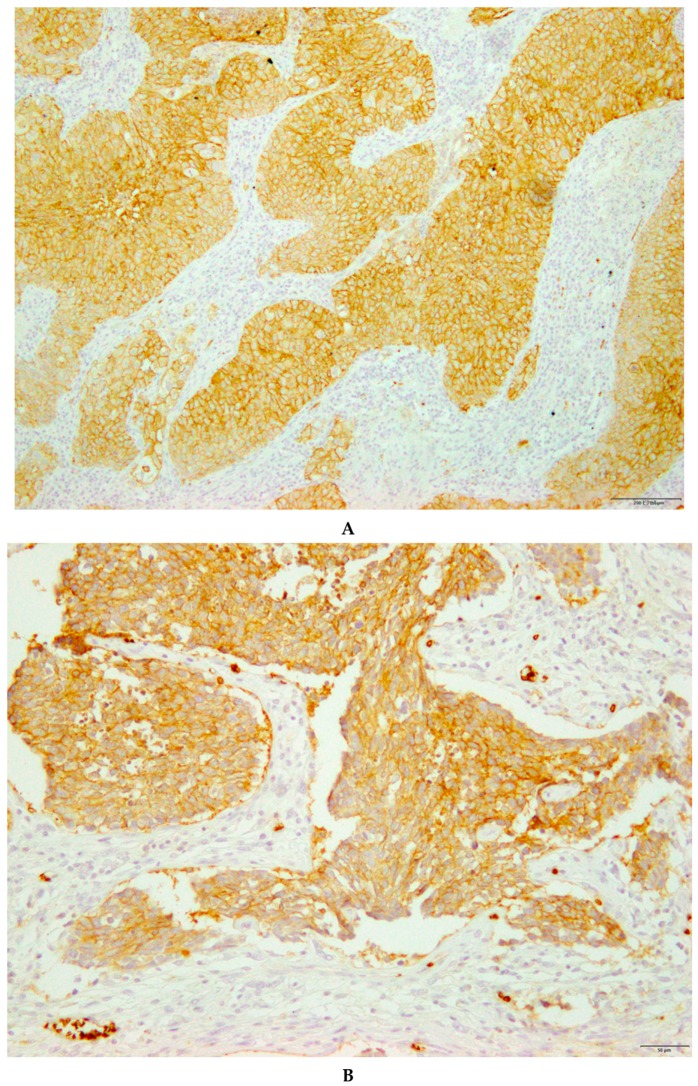
An 88-year-old male with PPC including a component of squamous cell carcinoma (**A**) GLUT1 was stained on the membrane of tumor cells, showing a score of 4. A 78-year-old female with PPC including components of squamous cell carcinoma and spindle cells: GLUT1 was stained throughout the squamous cell carcinomas (**B**) and partial lesions of spindle cells (**C**). A 77-year-old male with PPC including components of adenosquamous cell carcinoma and giant cells: GLUT1 was stained throughout the epithelial cells (**D**) and sarcomatous cells (**E**).

**Figure 2 jcm-09-00413-f002:**
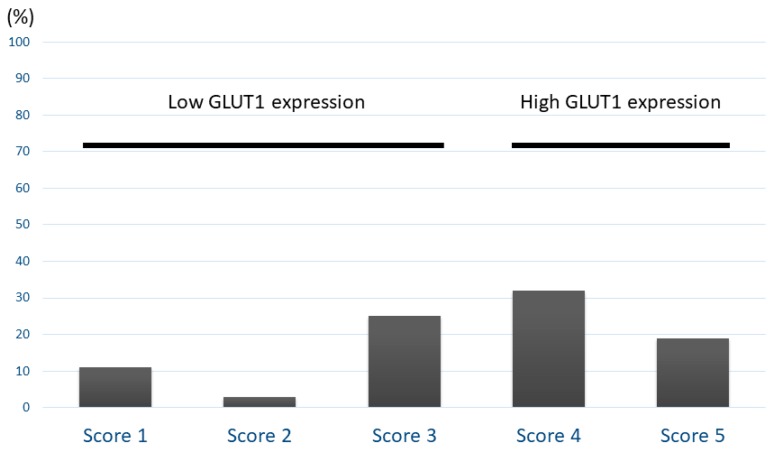
Distribution of GLUT1 expression according to scoring system. The frequencies of scores 1, 2, 3 4, and 5 for GLUT1 were 11%, 3%, 25%, 32%, and 19%, respectively.

**Figure 3 jcm-09-00413-f003:**
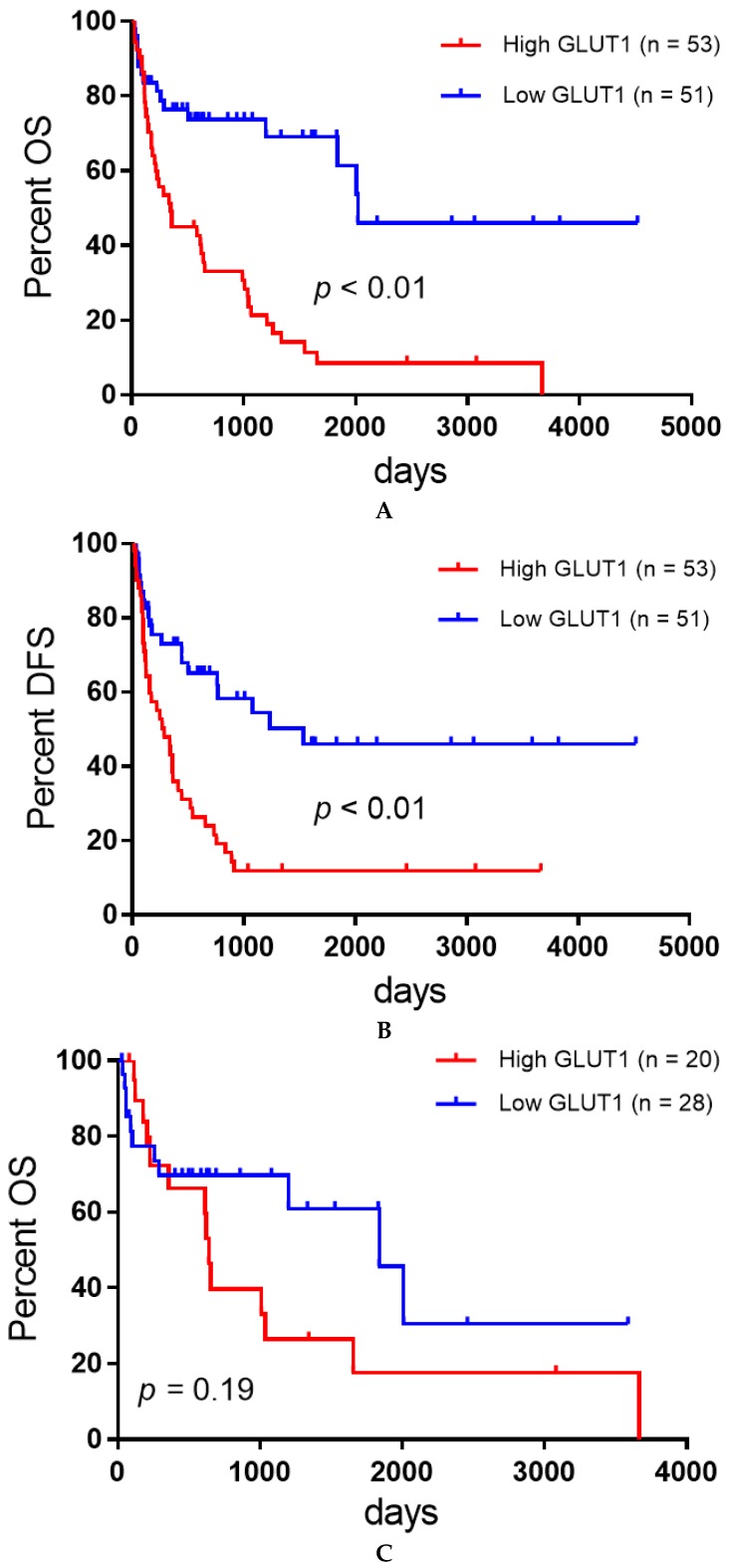
Kaplan-Meier survival curves for all patients (**A**,**B**), those with adenocarcinoma (**C**,**D**), and those with nonadenocarcinoma (**E**,**F**). Patients with high GLUT1 expression exhibited a significantly worse OS (A) and DFS (B) than those with low GLUT1 expression. No significant difference in the OS (C) and DFS (D) was observed between patients with adenocarcinoma with high and low GLUT1 expression, whereas the OS (E) and DFS (F) in patients with nonadenocarcinoma were significantly lower in those with high GLUT1 expression than in those low GLUT1 expression.

**Figure 4 jcm-09-00413-f004:**
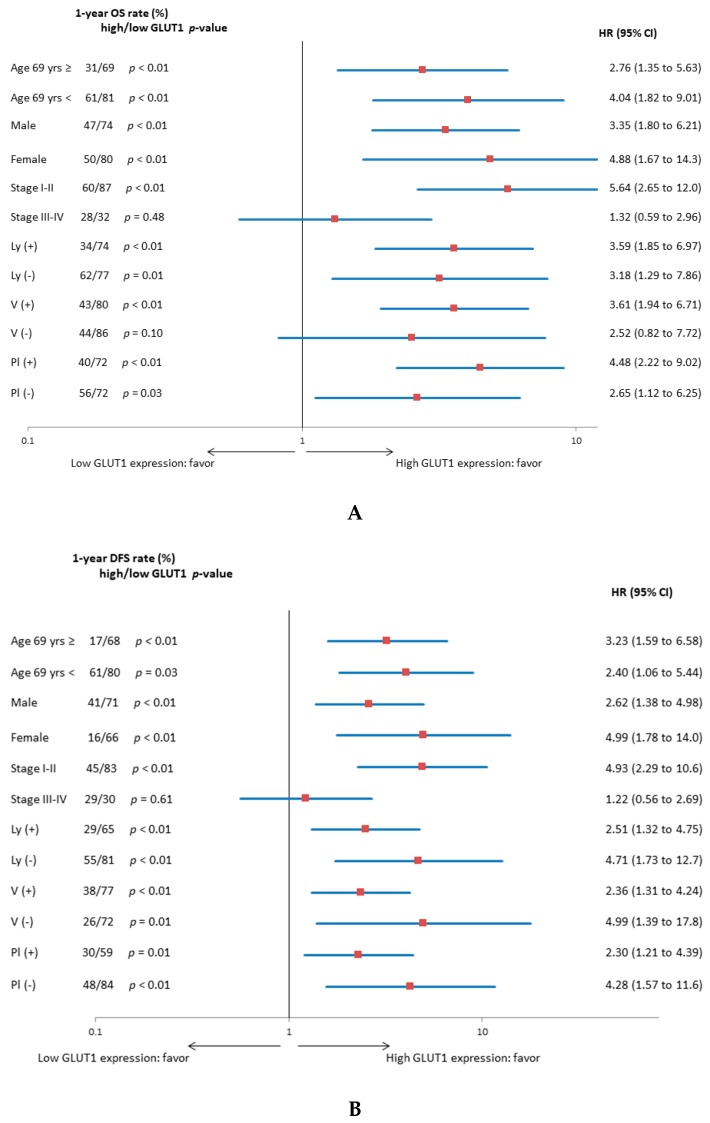
(**A**) Forest plot of one-year OS rate according to GLUT1 expression for each variable. (**B**) Forest plot of one-year DFS rate according to GLUT1 expression for each variable.

**Table 1 jcm-09-00413-t001:** Patient demographics according to GLUT1 expression.

Variables	GLUT1 Expression in All Patients			
	Total (n = 104)	High (n = 50)	Low (n = 54)	*p*-Value
**Age**				
**<69 years/≥69 years**	54/50	30/20	24/30	0.12
**Gender**				
**Male/Female**	79/25	35/15	44/10	0.25
**Smoking**				
**Yes/No**	84/20	40/10	44/10	>0.99
**T factor**				
**T1-2/T3-4**	65/39	25/25	40/14	0.11
**N factor**				
**Absent/Present**	72/32	33/17	39/15	0.53
**Stage**				
**I-II/III-IV**	69/35	28/22	41/13	0.03*
**Lymphatic permeation**				
**Absent/Present**	41/63	17/33	24/30	0.31
**Vascular invasion**				
**Absent/Present**	31/73	9/41	22/32	0.02*
**Pleural invasion**				
**Absent/Present**	48/56	17/33	31/13	<0.01*
**Adjuvant chemotherapy**				
**Absent/Present**	77/27	35/15	42/12	0.38
**Ki-67 labeling index**				
**High/Low**	50/54	31/19	19/35	<0.01*
*** <0.05**				

* *p* < 0.05 was considered statistically significant. *t*-test score was for continuous variables, and χ^2^ test for categorical variables.

**Table 2 jcm-09-00413-t002:** Univariate and multivariate survival analysis in all patients.

Variables	Overall survival (OS) in Total Patients				
	Univariate Analysis		Multivariate Analysis		
	1-Year Rate (%)	*p*-Value	HR	95% CI	*p*-Value
**Age (<69/≥69)**	48/73	0.41			
**Gender (female/male)**	60/58	0.66			
**p-stage (I-II/III-IV)**	75/29	<0.01*	1.53	1.21–2.12	<0.01*
**Ly (present/absent)**	52/71	0.21			
**v (present/absent)**	59/61	0.23			
**Pl (present/absent)**	53/66	0.07			
**Adjuvant CTx (present/absent)**	66/57	0.18			
**GLUT1 expression (high/low)**	43/75	<0.01*	1.72	1.29–2.34	<0.01*
**Ki-67 labeling index (high/low)**	60/61	0.77			
	**Disease-Free Survival (DFS) in Total Patients**				
**Age (<69/≥69)**	41/69	0.12			
**Gender (female/male)**	58/46	0.17			
**p-stage (I-II/III-IV)**	67/31	<0.01*	1.58	1.19–2.09	<0.01*
**ly (present/absent)**	47/68	0.04			
**v (present/absent)**	56/62	0.08			
**pl (present/absent)**	43/71	<0.01*	1.15	0.57–1.02	0.07
**Adjuvant CTx (present/absent)**	57/52	0.97			
**GLUT1 expression (high/low)**	36/72	<0.01*	1.44	1.08–1.95	0.01*
**Ki-67 labeling index (high/low)**	51/59	0.64			

CI = confidence interval; **p* < 0.05 is considered statistically significant, calculated with continuous variable; ly, lymphatic permeation; v, vascular invasion; pl, pleural invasion; GLUT1, glucose transporter 1; and HR, hazard ratio.

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
