# Peer review of "Prognostic Significance of Glucose Metabolism as GLUT1 in Patients with Pulmonary Pleomorphic Carcinoma"

_jcm, 2020, doi:10.3390/jcm9020413_

Round 1

Reviewer 1 Report

Major concern

The authors admit GLUT1 inhibitor is likely to suffer from toxicity problems which somewhat lowers the impact of the results. Nevertheless, the study add some support for attempting to target GLUT1 for PPC therapies. The authors demonstrates that Ki-67 labeling correlate with GLUT1 and GLUT1 serve as a significant independent marker for predicting poor prognosis for PPC. However, it is not clear from the manuscript what is the added benefit of GLUT1 as a prognostic marker compared to Ki-67 labeling or compared with current gold standards.

Minor:

1. L. 59 Please improve sentence: "cancers, and GLUT1 was identified an important marker for predicting poor outcome of patients with"

2. Table 2. Some times the authors use Glut1 versus GLUT1

3. Fig 3. E and F looks considerably more significant than A and B yet all figures are labelled with <0.01 which is not very accurate illustration.

4. Figure 3. y-axies annotation is confusing: "percent survival (PFS)" and "percent survival (OS)". "Percent PFS" and "Percent OS" would be better.

5. Figure 4. The Column labels on the left are misplaced making the figure hard to read. Also the first column is not defined.

Author Response

Comments and Suggestions for Authors

Major concern

The authors admit GLUT1 inhibitor is likely to suffer from toxicity problems which somewhat lowers the impact of the results. Nevertheless, the study add some support for attempting to target GLUT1 for PPC therapies. The authors demonstrates that Ki-67 labeling correlate with GLUT1 and GLUT1 serve as a significant independent marker for predicting poor prognosis for PPC. However, it is not clear from the manuscript what is the added benefit of GLUT1 as a prognostic marker compared to Ki-67 labeling or compared with current gold standards.

Re) Thank you for your generous comments. As you described, GLUT is necessary to survive in addition to the other nutrition such as amino acid and fatty. Out of different kinds of GLUT subtypes, it has been already described that GLUT1 and GLUT3 are highly expressed in cancer cells, according to previous reports. However, it is not elucidate whether the expression level of GLUT1 exhibits a close relationship with survival and tumor cell proliferation in patients with PPC. When discussing the prognostic factor by some biomarkers, we feel that the correlation with tumor cell proliferation should be investigated. Therefore, we examined the potential of GLUT1 as a prognostic marker compared to Ki-67.

Minor:

59 Please improve sentence: "cancers, and GLUT1 was identified an important marker for predicting poor outcome of patients with"

Re) According to reviewer’s comment, this sentence was revised.

Table 2. Some times the authors use Glut1 versus GLUT1

Re) According to reviewer’s comment, Glut1 was changed to GLUT1.

Fig 3. E and F looks considerably more significant than A and B yet all figures are labelled with <0.01 which is not very accurate illustration.

Re) As we calculated two decimal places, the number of p-value between E/F and A/B was similar.

Figure 3. y-axies annotation is confusing: "percent survival (PFS)" and "percent survival (OS)". "Percent PFS" and "Percent OS" would be better.

Re) According to reviewer’s comment, this annotation was revised.

Figure 4. The Column labels on the left are misplaced making the figure hard to read. Also the first column is not defined.

Re) According to reviewer’s comment, the column was revised.

Reviewer 2 Report

Imai et al. reported herein that the glucose transporter GLUT1 is a prognostic factor for predicting the poor prognosis of PPC patients.

The paper is well-written in general, and the data collection & statistical analysis are sounded. The conclusion is supported by the presented data. 

Author Response

Comments and Suggestions for Authors

Imai et al. reported herein that the glucose transporter GLUT1 is a prognostic factor for predicting the poor prognosis of PPC patients.

The paper is well-written in general, and the data collection & statistical analysis are sounded. The conclusion is supported by the presented data. 

Re) Thank you for your generous comments.

Reviewer 3 Report

Point 1. Overall the manuscript is well presented, however both the abstract and the patient demographics description (section 3.1) requires substantial editing.

Description of the results in the abstract are unclear and lack objectivity. The authors must consider revising the patient demographics description (section 3.1).

The authors describe a cohort of 104 patients, 29 of which harbored a combination of carcinomatous and sarcomatous components and in the remaining 75 primary tumours, carcinomatous components were identified in 48 patients with adenocarcinoma, 13 with squamous cell carcinoma, 8 with adenosquamous cell carcinoma, 2 with poorly differentiated carcinoma, and 4 with large cell carcinoma. Of the sarcomatous components, 69 patients exhibited spindle-cell type, 10 giant-cell type, and 25 both spindle- and giant-cell types.

Regarding this description is unclear the histology of each patient, which is crucial for the data presented in Figure 3 where patients are clustered into adenocarcinoma component and non-adenocarcinoma component. The authors must provide in supplementary information for each tumour sample the histological subtype and the correspondent carcinomatous and or sarcomatous component.

Point 2. The authors have used a quantitative Scoring Method (1, ≤10% of tumor area stained; 2, 11–25% stained; 3, 26–50% stained; 4, 51–75% stained; and 5, ≥76% stained) to access the expression of GLUT1 by immunohistochemistry. A more systematic approach must be used for the analysis of immunhistochemistry results. It is recommended the authors access 3 or more slides per patient sample, representative of the tumor anatomy. Authors must count the percentage of positive immunolabeled cells over the total cells in each selected area. The results must be presented as mean values of positively stained cells (and/or structures) among counted experimental groups with their standard deviations. This method can be automated with the use of special plugins for computer counting of general amount of cells and positively stained cells.

Point 3. Line 79 vs line 117. Different patient samples are reported.

Point 4. Line 243 – Please confirm the term adenocarcinoma in the sentence.

Author Response

Comments and Suggestions for Authors

Point 1. Overall the manuscript is well presented, however both the abstract and the patient demographics description (section 3.1) requires substantial editing.

Description of the results in the abstract are unclear and lack objectivity. The authors must consider revising the patient demographics description (section 3.1).

The authors describe a cohort of 104 patients, 29 of which harbored a combination of carcinomatous and sarcomatous components and in the remaining 75 primary tumours, carcinomatous components were identified in 48 patients with adenocarcinoma, 13 with squamous cell carcinoma, 8 with adenosquamous cell carcinoma, 2 with poorly differentiated carcinoma, and 4 with large cell carcinoma. Of the sarcomatous components, 69 patients exhibited spindle-cell type, 10 giant-cell type, and 25 both spindle- and giant-cell types.

Regarding this description is unclear the histology of each patient, which is crucial for the data presented in Figure 3 where patients are clustered into adenocarcinoma component and non-adenocarcinoma component. The authors must provide in supplementary information for each tumour sample the histological subtype and the correspondent carcinomatous and or sarcomatous component.

Re) According to reviewer’s suggestions, supplementary information as Figure S1 was provided in section 3.1. And we have revised abstract section.

Point 2. The authors have used a quantitative Scoring Method (1, ≤10% of tumor area stained; 2, 11–25% stained; 3, 26–50% stained; 4, 51–75% stained; and 5, ≥76% stained) to access the expression of GLUT1 by immunohistochemistry. A more systematic approach must be used for the analysis of immunhistochemistry results. It is recommended the authors access 3 or more slides per patient sample, representative of the tumor anatomy. Authors must count the percentage of positive immunolabeled cells over the total cells in each selected area. The results must be presented as mean values of positively stained cells (and/or structures) among counted experimental groups with their standard deviations. This method can be automated with the use of special plugins for computer counting of general amount of cells and positively stained cells.

Re) Thank you for your generous comments. There is no established method regarding the scoring system of GLUT1 expression within tumor cells. Thus, we always use the easy and simple scoring system like our study. Our method is supported by previous pathological reports. As you described, computer counting could be a promising and easy method. In this study, we feel that our method is appropriate for the evaluation of GLUT1 expression by immunohistochemistry.

Point 3. Line 79 vs line 117. Different patient samples are reported.

Re) We are sorry that 105 is mistyping, so, the number was corrected.

Point 4. Line 243 – Please confirm the term adenocarcinoma in the sentence.

Re) This sentence was corrected.

Round 2

Reviewer 3 Report

In Figure 3 is depicted Kaplan-Meier survival curves. Figure 3A-B is a representation of all patients, 3C-D of patients with adenocarcinoma and 3E-F of patients with non-adenocarcinoma. In Figures 3E-F, the overall survival rate and disease free survival shows the highest difference between low-GLUT and high-GLUT compared with Figures 3A-B and 3C-D.

These observations are not correctly reported in the text. Lines 208-211 and Lines 249-250.

The y-axis legend in Figure 3B, D and F must be corrected to DFS instead of PFS.

Author Response

Reviewer 3

Comments and Suggestions for Authors

In Figure 3 is depicted Kaplan-Meier survival curves. Figure 3A-B is a representation of all patients, 3C-D of patients with adenocarcinoma and 3E-F of patients with non-adenocarcinoma. In Figures 3E-F, the overall survival rate and disease free survival shows the highest difference between low-GLUT and high-GLUT compared with Figures 3A-B and 3C-D.

These observations are not correctly reported in the text. Lines 208-211 and Lines 249-250.

Re) We appreciate your pertinent observation. We have added and revised the following sentences to the revised manuscript as suggested: (lines 211-213 and lines 248-249): “We examined the prognostic significance of GLUT1 expression in patients with surgically resected PPC. We found that overexpression of GLUT1 is an independent factor for predicting poor outcomes and is useful as a prognostic marker in patients with non-AC component. In the patients with non-AC, OS and DFS showed the highest difference between low-GLUT1 and high-GLUT1 compared with all patients and subgroup patients with AC component.” And “Therefore, it may be difficult to administer inhibitors of GLUT1 as treatment for PPC with non-AC component in clinical practice.”.

The y-axis legend in Figure 3B, D and F must be corrected to DFS instead of PFS.

Re) Thank you for your suggestion. According to reviewer’s comment, the y-axis legend in Figure 3B, D and F was revised.